# Observational Study Exploring the Efficacy and Effectiveness of a New Model of Peer-Assisted Simulation-Based Learning Clinical Placement

**DOI:** 10.3390/ijerph19084505

**Published:** 2022-04-08

**Authors:** Diane Dennis, Lora Cipriano, Ginny Mulvey, Stephanie Parkinson, Alan Reubenson, Anne Furness

**Affiliations:** 1Curtin School of Allied Health, Curtin University, Perth 6845, Australia; lora.cipriano@health.wa.gov.au (L.C.); g.mulvey@curtin.edu.au (G.M.); s.parkinson@curtin.edu.au (S.P.); a.reubenson@curtin.edu.au (A.R.); a.furness@curtin.edu.au (A.F.); 2Sir Charles Gairdner Hospital, Nedlands 6009, Australia

**Keywords:** peer-assisted learning, physiotherapy, simulated patients, simulation

## Abstract

(1) Background: Immersive simulation-based learning is relevant and effective in health care professional pre-licensure training. Peer-assisted learning has reciprocal benefit for the learner and the teacher. A fully simulated model of fieldwork placement has been utilised at Curtin University since 2014, historically employing full-time faculty supervisors. Due to the COVID-19 pandemic in 2020, traditional clinical placement availability diminished. (2) Methods: This mixed-methods prospective observational study aimed to translate the existing faculty-led placement for penultimate-year physiotherapy students to a peer-taught model, thereby creating new teaching placements for final-year students. Final- and penultimate-year physiotherapy students undertook the fully simulated fieldwork placement either as peer learners or peer teachers. The placement was then evaluated using four outcome measures: The ‘measure of quality of giving feedback scale’ (MQF) was used to assess peer learner satisfaction with peer-teacher supervision; plus/delta reflections were provided by peer teachers and faculty supervisors; student pass/fail rates for the penultimate-year physiotherapy students. (3) Results: For 10 weeks during November and December 2020, 195 students and 19 faculty participated in the placement. Mean MQF scores ranged from 6.4 (SD 0.86) to 6.8 (SD) out of 7; qualitative data reflected positive and negative aspects of the experience. There was a 4% fail rate for penultimate-year students for the placement. Results suggested that peer learners perceived peer-led feedback was of a high quality; there were both positives and challenges experienced using the model. (4) Conclusions: Physiotherapy students effectively adopted a peer-taught fully simulated fieldwork placement model with minimal faculty supervision, and comparable clinical competency outcomes.

## 1. Introduction

Between 2013 and 2015, Health Workforce Australia (HWA) funded a national project embedding simulation as a learning and teaching modality across physiotherapy clinical fieldwork programs throughout Australia [1]. These involved students working face to face with professional actors who enacted scripted simulated patient (SP) roles in novel scenarios and bespoke life-like environments, rather than real patients in real clinics. At Curtin University, the clinical placement ran for a total of 18 days, with students spending 6 days working with a range of SPs who presented with conditions representative of each of the three core physiotherapy practice areas—cardiopulmonary, neurosciences and musculoskeletal [2]. All students were supervised by full-time clinical tutors for the duration of the placement in a 1:4 model. The SPs were also full time, participating in both the face-to-face clinical scenario with students as well as the debrief of student performance. The goals of the placement related more to the human factors around health care service delivery than clinical acumen. That is, students were expected to primarily gain experience in tasks such as history taking, developing rapport, note reading and writing, handover, teamwork, leadership and time management, all in the context of ‘being’ a physiotherapist, rather than necessarily the clinical skills related to the profession itself. It was considered a preparatory adjunct to ‘real’ clinical placements that take place in the final year of the program.

Although the initial funded model was costly, the university recognized the merit of the placement in terms of student outcomes that were found to be comparable to traditional ‘real’ preparatory placements [2] and the growing concern around the lack of traditional fieldwork placement options for an increasing number of students. From 2014, within the resource restraints of limited space, staff and money, the university continued to implement a derivation of the original model, whereby approximately half the cohort (*n* = 96) of physiotherapy students in their penultimate year of study (PYPT) undertook a fully simulated experience annually in two 18 day blocks, with the remaining students continuing to undertake traditional ‘real’ clinical placements. The derived simulation model was streamlined to gain maximal efficiencies in the utilisation of actor and supervisor time. Supervisors attended the placement full time with a 1:8 student ratio. The SPs continued to be trained according to best-practice standards for physiotherapy [3,4,5]; however, their time onsite each day was rationalized to the duration of the clinical encounter, rather than including additional student debrief time. Students encountered approximately 30 simulated patients in high-fidelity hospital and outpatient settings over 18 days. When they were not interacting with SPs, they were occupied completing novel self-directed workbook tasks related to either specific SPs they had encountered, or to the clinical reasoning or skillset that was required for that core stream. Examples of these tasks included writing case notes; constructing assessment or treatment plans; completing discharge summaries and letters to other professionals such as General Practitioners; reviewing chest X-rays and arterial blood gas results; and practicing pre-operative surgical education using role-play.

Between 2015 and 2019, anecdotal feedback from both students and faculty suggested that despite the worthiness of a ‘real’ traditional placement, there was a perception of inequity in that all PYPT students were not presented the same opportunity to undertake the fully simulated placement, which had become highly valued among the cohort. With the advent of the COVID-19 pandemic in 2020, and the limited supply of ‘real’ fieldwork opportunities, there became an increased urgency to create a simulation-based model that incorporated the entire PYPT student cohort in a cost-effective way. During the pandemic, this cohort was not the only group impacted by the paucity of traditional fieldwork placements. Physiotherapy students in their final year of study (FYPT) faced delayed graduation with incomplete clinical training.

There is significant literature supporting the peer-assisted learning (PAL) model in health care professional training, with evidence suggesting bilateral benefit for both the learners and teachers [6,7,8,9,10,11,12]. Peer learners experience less fear of judgment and are more comfortable, with less attachment to assessment [13]. Having freshly mastered the content themselves, peer teachers were more confident [14] and relate better to the material and therefore pitch it at a more appropriate level [15]. Literature specifically related to peer learning in physiotherapy curricula is emerging and appears positive [16,17,18] especially as it pertains to the comparative faculty-led teaching [16]. Additionally, combining simulation and PAL as two learning and teaching approaches is not new. In fact, PAL in simulation training has been found to be a feasible low-cost alternative for health care professional education [19,20].

Given that ‘teaching’ is a required physiotherapy graduate competency, a mutual and innovative solution was for us to adapt the existing faculty-led model to a near-peer-teaching model, thereby increasing the capacity of our program to include all PYPT students whilst at the same time creating additional FYPT ‘learning and teaching’ fieldwork placements. This innovation addressed the workforce resource limitation. The workspace limitation was addressed by the coincidental availability of an additional new simulation facility that became accessible in 2020. This building doubled the previous capacity, having four bespoke rooms that could each be set up with two hospital beds and other medical paraphernalia (such as oxygen and suction) to mimic the patient rooms of an acute hospital. It had another four rooms with patient plinths that could be set up as rooms in an outpatient clinic area. With careful scheduling, these facilities could accommodate more than 50 students to undertake simulation-based activities simultaneously.

The aim of this study was to translate the existing full-time faculty-led simulation-based clinical placement for PYPT physiotherapy students to a PAL model, and to assess the efficacy and effectiveness of the PAL model from the perspective of peer learners, peer teachers and the faculty supervisors of the peer teachers.

## 2. Materials and Methods

This was a mixed-methods prospective observational study involving the development and implementation of a new model of the simulation-based clinical placement. Following Curtin University Human Research and Ethics committee approval (HRE2020-73659), the target population was identified from three groups:All PYPT students. Those students enrolled in Integrated Clinical Science (PATH3002) and Clinical Science Fundamentals (PATH6006) were invited to participate in this study as peer learners.FYPT students enrolled in Physiotherapy Clinics 5 (PHTY4015) were selected to be peer teachers based on their need to complete a fieldwork placement in order to graduate. To be eligible, they were required to have completed successfully either the faculty-led simulation placement during their own penultimate year of study, or at least one clinical placement in each of the core areas of practice (cardiopulmonary, neurosciences and musculoskeletal). Final-year physiotherapy students who had failed any of these placements were excluded from participating as peer teachers.Over the duration of the clinical placements, 17 peer-teacher supervisors participated in the clinical and all were invited to participate in this study. Each was considered a clinical expert in their area of practice (cardiopulmonary, neurosciences or musculoskeletal) and had considerable experience supervising students on traditional fieldwork placements.

### 2.1. Development of the Model

A new model of supervision (see Figure 1) was developed such that from 8:30 am to 4:30 pm every weekday, up to 12 FYPT students were responsible for supervising up to 96 PYPT students (ratio 1 FYPT:8 PYPT students). This was the same ratio that had existed in the faculty-led model. An additional FYPT student was allocated to each placement in a ‘roving’ role, whereby they provided extra support to meet the demands of students performing at either tails (both poor and exceptional students). These students undertook additional one-to-one tutoring and feedback, on the request of either the site supervisor or their FYPT peers. They also provided periods of teaching relief for FYPT student peer teachers when needed; covered any FYPT student or SP absences; and assisted with overall simulation scenario setups.

To support peer teachers, each was assigned a faculty supervisor with whom they were required to liaise with either by telephone or online for 30 min at the end of the day, Mondays to Thursdays. This support took the form of sharing specific expert knowledge related to the core stream (cardiopulmonary, neurosciences or musculoskeletal) as well as advice around clinical supervision of the PYPT students. On Fridays, these faculty supervisors attended the clinic in person each week to oversee PYPT performance and FYPT student assessment of that performance, for which the FYPT students held primary responsibility. 

The placement was implemented over two sites simultaneously, and assessment of the FYPT student performance was undertaken by two additional faculty supervisors who each remained full time at one site for the duration of the placement. These supervisors were each responsible for up to 7 FYPT students, observing their performance as peer teachers, and providing daily feedback as to performance, with titration and remediation where required.

Assessment of all students was unchanged from previous traditional and simulation placements, whereby an iteration of the Assessment of Physiotherapy Practice (APP) rubric [21] was utilised, determining a pass or fail score. Each Friday, PYPT students received formal feedback from their assigned FYPT student related to their performance in the stream they had just completed, with an overall summative assessment provided on the final Friday. Both Faculty peer-teacher supervisors and the site supervisor provided oversight for all assessments.

Four measures were used to evaluate the fully simulated placement:Penultimate-year placement fail rate of both student cohorts was collected and compared to 2019 data, where the student fail rate for the simulated placement was 9 percent.The quality of the teaching and feedback received from their peer teacher is most important for peer learners [17]. Accordingly, peer-learner students completed the ‘measure of quality of giving feedback scale’ [22]. This is a validated 20-question survey that takes less than 10 min to complete, where peer learners rated the quality of the feedback provided by their peer teacher on a 7-point scale (1 = not at all to 7 = very well). The survey was delivered to each peer learner via email using Qualtrics software (Qualtrics, version January 2021, Provo, UT, USA. https://www.qualtrics.com accessed on 8 April 2022) at the end of each week.The most important thing to capture from the peer teacher’s perspective was their impression of the PAL model, particularly in the context of their previous experiences of traditional fieldwork placements. Accordingly, peer teachers wrote plus/delta reflections of their perception of positive and negative aspects of the model at the end of their five-week placement.As personnel with significant clinical supervision experience, faculty supervisors wrote plus/delta reflections of their perception of positive and negative aspects of the new PAL model at the end of their period of supervision.

### 2.2. Implementation of the Placement

#### 2.2.1. Allocation of Students to Placement

All PYPT students were allocated to one of two consecutive 3 week placements. The FYPT students were allocated to one of two 5 week placements that overlapped with each other for the purpose of the in situ training of the second FYPT student cohort (see Table 1).

#### 2.2.2. Training and Upskilling of FYPT Students to Be Peer Teachers

All FYPT students received preparatory face-to-face in-person training to participate as peer teachers during the placement. The objectives of this training were that they:Understood the PAL approach; simulation as a learning and teaching modality; and the bespoke model of the placement in terms of its structure; the lines of responsibility, reporting and communication; and outcomes.Were familiar with the 30 simulation scenarios, including the expectations of the SP in enacting their role.Understood the expectations for PYPT student performance and were confident in assessing this with guidance from their faculty supervisor using the rubric provided.Were confident and practiced in providing constructive feedback during the placement and final assessment findings to their students.

Training included implementation of a workbook which outlined the learning objectives and expected performance levels of PYPT students for each case, as well as practical upskilling in the delivery of the scenario and debriefing practice. Bespoke training videos were also recorded and utilised to replicate the ‘look’ of the SPs within the scenario, and best-practice debriefing.

This training was implemented in two different ways: the first FYPT student group completed 3 days of didactic instruction and practice using roleplay and coaching from faculty supervisors, without PYPT students or SPs. The second group trained over 6 days (during their first 2 weeks) by observing the SPs in situ with the first group of FYPT students supervising their PYPT students.

In addition to the FYPT and PYPT student peer-assisted relationship, during their final week, first placement FYPT students were also able to observe, assist and coach the second cohort of FYPT students supervised during their first week (see Table 1).

#### 2.2.3. Training and Upskilling of Faculty Supervisors

All faculty supervisors were schooled in the new framework in a face-to-face in-person workshop, where the new model and the roles and responsibilities and outcome measures were explicitly outlined. All supervisors received a workbook outlining the cases and the expectations around both the students and the performance of the SPs. Most of these personnel (*n =* 14, 82%) had supervised previous iterations of the simulated placement between 2015 and 2019.

#### 2.2.4. Orientation and Consent to Participate in This Study

Each group (peer learners, peer teachers, and faculty supervisors) was oriented to the organisation, structure, and procedures of the placement in separate orientation sessions. During these meetings, students and staff were provided a participant information sheet as to this study, and only data from those providing written consent were included in the dataset.

#### 2.2.5. Data Collection Timepoints

At the end of each of the three weeks of their placement, all PYPT students completed the ‘measure of quality of giving feedback scale’ related to the performance of their supervisor as a peer teacher; At the end of their five-week placement, all FYPT students completed a written reflection of their experiences within the PAL model; at the end of their clinical supervision, faculty supervisors provided their written reflections.

### 2.3. Data Analysis

Quantitative survey data were exported and collated from Qualtrics software (Qualtrics, version January 2021, Provo, UT, USA. https://www.qualtrics.com accessed on 8 April 2022) into an Excel spreadsheet (A.F.), analyzed (D.D.), and presented as median scores [IQR]. Thematic analysis of the qualitative dataset [23] was undertaken by four investigators (D.D., L.C., A.R. and G.M.) who independently coded data deductively using a ‘Framework Analysis’ methodology and a post-positivist approach [24]. Emergent themes and subthemes that fit the broad pattern of meaning were then identified, discussed and agreed upon.

## 3. Results

During November and December 2020, 171 PYPT students undertook the fully simulated placement in one of two 3 week blocks, 170 of whom consented to participate in this study. Over the same period, 24 FYPT students undertook clinical supervision as peer teachers, all of whom consented to participate in this study. There were 17 peer-teacher supervisors who provided teaching and learning feedback and expertise specific to one of the core streams (either cardiopulmonary, neurosciences or musculoskeletal). Of these, 16 consented to participate in this study.

The final placement fail rate reflected results consistent with previous years, whereby seven PYPT students failed, representing a failure rate of 4 percent of the total PYPT student cohort. All FYPT students passed the placement.

Peer learners’ ratings of feedback received from peer teachers from the measure of quality of giving feedback scale are shown in Figure 2, with mean scores ranging from 6.4 (SD 0.86) to 6.8 (SD 0.47). No rating below 6 for any domain suggests that the peer learners’ overall perception of the quality of the feedback they received from peer teachers was high.

A summary of the themes emerging from qualitative data provided by peer teachers (*n* = 16) pertaining to the positive elements of the placement from their perspective are shown in Table 2.

These components included the personal experience encountered by peer teachers, whereby enjoyment, appreciation, and confidence were expressed.


*“This placement was challenging but greatly enjoyable. It was my personal favorite placement and I felt the utmost support from the students, my peers, and the (de-identified for blind review) and primary supervisors.”*



*“In the scenario in which I have a student to supervise in the future, this placement will have been an irreplaceable experience to build confidence and exposure to supervising.”*


Deepening of skillsets was described including collaboration, clinical skills, professional and organisational skills, and interpersonal skills.


*“Having input from various sources and varying opinions I believe helped myself and the students have a greater learning experience.”*



*“I have been able to learn about the cases and different ways to approach/treat through the peer learning students e.g., understanding their clinical reasoning as to why they are performing a particular assessment/doing a task in a specific position and then being able to add that to my approach with patients.”*


There was also acknowledgment of the attributes of using PAL and simulation as the teaching models for the placement. Subthemes relating to PAL included peer-to-peer relatability and approachability:


*“In summary, I believe the standout benefit and positive aspects of the way in which this placement runs would be providing the students with peer-teachers who have recently done their clinical placements, are at a similar stage of learning with only one year additional experience, and are in a similar age bracket. I believe this facilitated their comfort within sim as they were provided with additional empathy and understanding of their stress and how they’re feeling, and encouragement to use the sim as a safe place to make mistakes and to trust themselves and develop their clinical reasoning skills. They were also provided with frequent examples as to how these skills were relevant in fourth year and our personal recent experiences with these situations.”*



*“In terms of the peer learners, my group reported learning a lot over the three weeks. I believe the framework provides them with a great experience to seek learning, whereby students (especially the quieter or weaker students) appeared less daunted to ask questions in the peer teacher model.”*


These observations triangulate with several of the highest scoring items of the measure of quality of giving feedback scale such as establishing a climate of trust and support (mean 6.7; SD 0.55); using non-judgmental language (mean 6.8; SD 0.47); and allowed students time to self-assess (mean 6.7; SD 0.54).

Mentorship and the autonomy and flexibility of the model were also described as being positive.


*“Being able to have a mentor relationship with each stream supervisor to discuss other methods of approaching patients or ask any questions that came up and I wasn’t able to answer.”*


The opportunity for reflective practice was identified as an unanticipated constructive facet, and the authenticity of the simulated environment was described as helpful in having students immerse in the experience.


*“This model of peer teaching demands self-reflective practice from students and in doing so encourages us as mentors to do the same. I believe self-reflection is an essential component of learning which has facilitated personal growth in communication, particularly within a group of various personalities as well as improving my teaching/educational skills.”*



*“Simulation environment was challenging for the students however created real life scenarios which would benefit them for practical placements it also allowed me to refine my skills further throughout (my) final year.”*


A summary of the themes emerging from qualitative data provided by peer teachers (*n =* 16) pertaining to the challenging elements of the placement from their perspective are shown in Table 3.

The hierarchy within the structure of the placement provided some challenges for the peer teachers. There were some difficulties experienced when faculty supervisors questioned the knowledge of FYPT students in front of PYPT students; there was also sometimes complexity in the relationship between peer teachers and peer learners in terms of maintaining authority and an appropriate level of respect. Pre-existing relationships compounded these difficulties.


*“Establishing a difference between the year groups… this issue was amplified with mature age students or students who are older than their supervisor, however these are situations which are also encountered on fourth year prac for mature aged students and they will need to be given and learn strategies as to how they can manage their emotions regarding this and show respect towards those with more physiotherapy experience.”*



*“Having friends previously in our year group that you are supervising and having to try to create a professional boundary and provide constructive feedback.”*


Peer teachers acknowledged that a lot of preparation was required for the cases, and this was at times challenging. They also identified that mentoring high-performing students was sometimes demanding when they did not feel as comfortable with their own level of understanding.


*“It was the hardest clinical placement I undertook in terms of stretching my knowledge.”*



*“I felt a lot of responsibility for their performance when in reality the students had not prepared adequately and were not performing well.”*



*“Trying to create challenging situations for the more advanced peer-learners who are finding simulation scenarios simple.”*


Some students did not anticipate the non-clinical interpersonal issues that might arise during fieldwork placement, such as interpersonal problems, and found these difficult. Time management was another peer teacher-centered issue.


*“Took some time to realize that supervising is largely managing/handling personal and interpersonal issues (such as students taking feedback personally).”*



*“I found it difficult for myself to ensure I was giving the student the best opportunity to pass the placement and improve, however I found it hard to also spend just as much time with my other students who were performing really well.”*


A few students identified a lack of buy-in from peer learners to the PAL model and this was problematic.


*“I believe some PYPT students found the peer-supervisor role difficult and found it challenging to accept feedback from a fellow peer. I can appreciate this viewpoint but found it difficult at times to work with certain students who had less confidence in the peer-teacher model. It was difficult to provide feedback and assist some students in developing their clinical abilities when the receptiveness to feedback was lacking due to a lack of confidence in my role as a supervisor. I also found it difficult at times treading the line of “peer” and “supervisor.”*


Delivering feedback was the final challenging theme to emerge. This included the ability to articulate bespoke, succinct specific feedback while navigating individual differences. Having difficult conversations around poor performance was particularly tough.


*“Learning to provide non-repetitive, summarized, and specific feedback was very difficult.”*



*“Having to provide individualized feedback to students (especially those performing well).”*



*“Having to feedback to a student when they were non-competent. Admittedly I found this to be particularly difficult when I was having to provide that feedback by myself and when students became emotional secondary to that feedback. I am a very empathetic and emotional person, and this amplified how difficult the feedback sessions could be, however this is also a reflection on how I could develop my resilience.”*


Four themes arose from the feedback from supervisors of the peer teachers, and these are shown in Table 4. The first feedback theme was the PAL model itself. It was recognized as enhancing collaborative learning and providing a less stressful environment for peer learners.


*“My observation was that some PYPT students were possibly more comfortable asking questions of the FYPT peer supervisors than they may have been asking questions of qualified clinicians.”*


This was sometimes felt to be to the detriment of the experience, whereby PYPT students did not take the FYPT student feedback as seriously as they might have from a faculty supervisor. As identified by the peer teachers, issues around the hierarchy were also recognized by the peer-teacher supervisors, especially where friends were supervising friends.


*“Students tend to not be as forthcoming with constructive criticism—were ‘too nice’ especially to friends.”*


There was an appreciation that peer teachers were afforded new insights into the challenges of student supervision, while at the same time managing to navigate their learners toward competency and successful completion of the placement.


*“It was good to see how he developed over the prac and to hear his insights into the challenges that clinical supervision involves. I think these insights will serve him well as he starts to work as a new-graduate clinician.”*


The impact of the PAL model on the faculty themselves was another subtheme, whereby a lighter supervisory role was appreciated at a time where staffing levels were stretched with the impact of the COVID-19 pandemic.


*“It significantly lightened the load for me, which is important when staffing is getting ever more stretched.”*


The use of simulation as a teaching modality was the second theme to emerge. Although most peer-teacher supervisors had experienced the utility of simulation, the value of the immersive environments was identified, as well as the importance of quality actors.


*“Being a simulated situation, it was easy to time out and in as required which is a great safety feature. Having said that it wasn’t something that was required too often.”*


One respondent suggested that actors might have a teaching role in future iterations of the placement by providing their own feedback as to student performance during scenarios. There were also comments as to the overall positive value that the simulated placement offered in preparing students for their final year of clinical training. The caveat was that those FYPT peer teachers who used the modality were good teachers themselves and had sound knowledge of the subject area.


*“My impression was that the placement served its purpose well. As an ex manager I was confident that the students who had passed were well prepared to undertake their clinical placements in a hospital environment. Whilst clearly not at ‘expert’ level they were able to demonstrate safe clinical practice and come up with at least a basic intervention. Is clearly an ideal building block to prepare them for their final year of physio.”*


The support provided to everyone involved in the placement was the third theme of peer-teacher supervisor feedback. This included the training and preparation received before and during the placement; the value of the daily support delivered by faculty—although 30 min each day was felt to be short in some cases; and the support that the roving FYPT provided.


*“(There was) very good clear and specific training of FYPT students prior their supervision of their PYPT student group.”*


The final peer-teacher supervisor feedback theme centered on the FYPT students themselves. Subthemes included their level of enthusiasm, organisation and diligence, confidence related to their performance and competence as imminent graduates; and their teaching acumen. There was a sense that the FYPT students consolidated their learning with a deeper understanding of the clinical scenarios during the placement. There was a perception of a great variation in the teaching competencies among the FYPT student cohorts, including the extent to which they were able to effectively convey information during the daily feedback conversations.


*“The responsibility of supervision definitely prompted the FYPT peer supervisors to revise their knowledge. I would suspect that had they been on a ‘regular’ clinical placement, their preparation would have been significantly less! My impression is that their revision of knowledge was internally motivated by not wishing to look foolish when asked questions by the PYPT students. Whereas on a regular placement, a significant proportion of students are relatively passive in their preparation and only revise knowledge when prompted by the external motivator of being asked to do so by a supervising clinician. I know that without doubt, the FYPT student that I was supervising worked exceptionally hard during this prac. He spent a lot of time reviewing previous learned material and was keen to discuss and clarify concepts with me.”*



*“They sometimes lacked a deeper understanding of some of the clinical presentations but were well able to supervise and teach very well at the PYPT students level required.”*



*“When it came to giving feedback to the PYPT students I was impressed with just how well the FYPT students did this. They were very much on the mark with their assessments. Even though we gave feedback to the students together I took a very back seat role.”*


## 4. Discussion

The overall success of this novel peer-assisted clinical placement model is reflected in three summary comments. One member of peer-teacher supervising faculty said:


*“This was a great set up! All parties get something positive out of it. It creates meaningful placements for both PYPT students and FYPT students.”*


Another said:


*“I think it was a fantastic development in a difficult year. Amazing achievement by all staff involved in its planning and execution.”*


A FYPT student commented: 


*“The near peer teaching method utilised in this simulation model approach was overall a great success.”*


All statements were supported by the qualitative analysis of peer teacher, peer learner and peer peer-teacher supervisor outcome data, which in turn triangulated nicely with the quantitative data related to the high quality of the feedback provided to PYPT students by FYPT students. The overall pass rate of 96% for PYPT students and 100% for FYPT students, which was comparable to previous iterations of the placement, further reflects a positive outcome from the novel model, and is supported in the literature, where it has been reported that peer teaching in highly selective contexts achieves learner outcomes that are comparable with those produced by faculty-based teaching [25].

In summary, these data demonstrate that FYPT student peers who are specifically targeted as competent students in their own fieldwork placements have the capacity to deliver quality instruction to their PYPT student peers. In addition, they report enjoyment and benefit from the experience of doing so.

One of the positive attributes of the teaching model identified by peer teachers was the opportunity to self-reflect on performance. Self-reflection is an important skill for becoming an independent learner [26], and the simulated clinical encounters offered peer learners progressive independence such that peer teachers were able to coach them without the constant formal guidance of a faculty supervisor.

Some peer teachers found supporting high-performing students to be difficult; however, the support provided by faculty-supervisors enabled more advanced questions and scenarios to be answered and addressed to optimise all students’ learning. At the other end of the spectrum, peer teachers also found supervision of poorly performing peer learners to be challenging. This was compounded when the peer learners were seen to be trying hard to master content but failing to do so. The event of having to plan and execute a ‘difficult’ conversation was taxing, but a worthy learning experience.

One of the biggest challenges to emerge from participants’ reflections that requires ongoing consideration for this model to be successful is that of pre-existing relationships within the peer learner/teacher hierarchy. One example was where a PYPT student had failed or deferred their study such that they were once part of the FYPT student cohort and were now being supervised by a FYPT student ‘friend’. This led to some difficult situations, where there may have been an expectation on the part of the peer learner for lower accountability and the potential for compromised rigor in final peer assessment. The influence, support and scrutiny provided by the faculty supervisors overcame this challenge from an institutional perspective, whereby no PYPT students passed when they should have failed. The impact this had on the peer teachers who failed their friends, both personally and in relation to ongoing relationships post-placement, was beyond the scope of this study and not measured. Having had this situation arise in the first 3 week placement, all FYPT students teaching the second 3 week placement were asked to peruse their student list and declare any conflict of interests ahead of time, and students were re-allocated such that prior relationships did not impact either learner or teacher.

Three reasons cited in the literature as an indication for the use of a peer-teaching model in medicine are that it alleviates faculty teaching burden; enhances intrinsic motivation; and prepares physicians for their future role as educators [27]. Each of these reasons was supported by the peer-teacher supervisors’ reflections.

## 5. Limitations

A strength of this study was the high responder rate from participants in the project (100% of peer learners; 58% of peer teachers; and 94% of faculty supervisors). The main limitations of this study are that it reports data from only one professional group (physiotherapists) following one iteration of the fieldwork placement model. The extent to which results may be generalized to other professions is therefore unknown. Further limitations relate specifically to either the simulation or peer-teaching models. In terms of simulation, physiotherapists graduate based on clinical competencies rather than number of clinical hours completed, and there is no limit to how many clinical hours may be undertaken using simulation. This is not the case with other professions, including nursing, such that translation of the entire model to other professions may be restricted. The peer-assisted learning model may also present constraints for some professions in terms of course recognition and accreditation. Exercise and Sports Science Australia for example stipulate that only qualified exercise physiologists can oversee the clinical fieldwork of pre-licensure students, which presents restrictions for the model.

## 6. Conclusions

Faced with uncertainty related to the availability of clinical placements with the advent of COVID-19, our team translated an existing 3 week fully simulated clinical placement previously taught by faculty to half of our PYPT students to one that was taught by FYPT students using a PAL model to our entire PYPT cohort. This involved significant training of students and supervisors, and the utility of a new simulation facility that provided an increased physical capacity to deliver the simulation scenarios en masse.

Penultimate-year physiotherapy students achieved comparable pass/fail placement outcomes and rated the feedback provided by their FYPT teachers positively. Final-year students who themselves achieved a 100% pass rate for the placement reported that it was one of the hardest yet most satisfying clinical placements of their final year. They reported the value of the placement as a ‘capstone’ event, in that it was incumbent on them to review their entire coursework in order to teach the PYPT students, which in turn became a positive and useful segue to enter the profession as new graduates. Qualitative data from peer teachers, peer learners and the staff that supervised two iterations of the model over 6 weeks were unanimously positive. This model has increased the capacity of our program to deliver worthy fieldwork placements for both FYPT and PYPT students in the face of limitations due to the COVID-19 pandemic.

It is important to understand that although clinical skill and background knowledge are considered, the primary goals of the PYPT student placement relate more to the human interaction associated with the delivery of health care rather than the high level of clinical acumen that is the expectation of both FYPT students on ‘real’ clinical placement and successful graduates. We do not advocate for this model of ‘simulated’ clinical placement to replace existing ‘real-life’ FYPT clinical placements; rather, it can be utilised for students in their penultimate year of training as a bridge to ‘real’ clinical placements. That said, the current model also provides a capstone clinical placement for those FYPT students who act as peer teachers at the end of their clinical program prior to completion.

## Figures and Tables

**Figure 1 ijerph-19-04505-f001:**
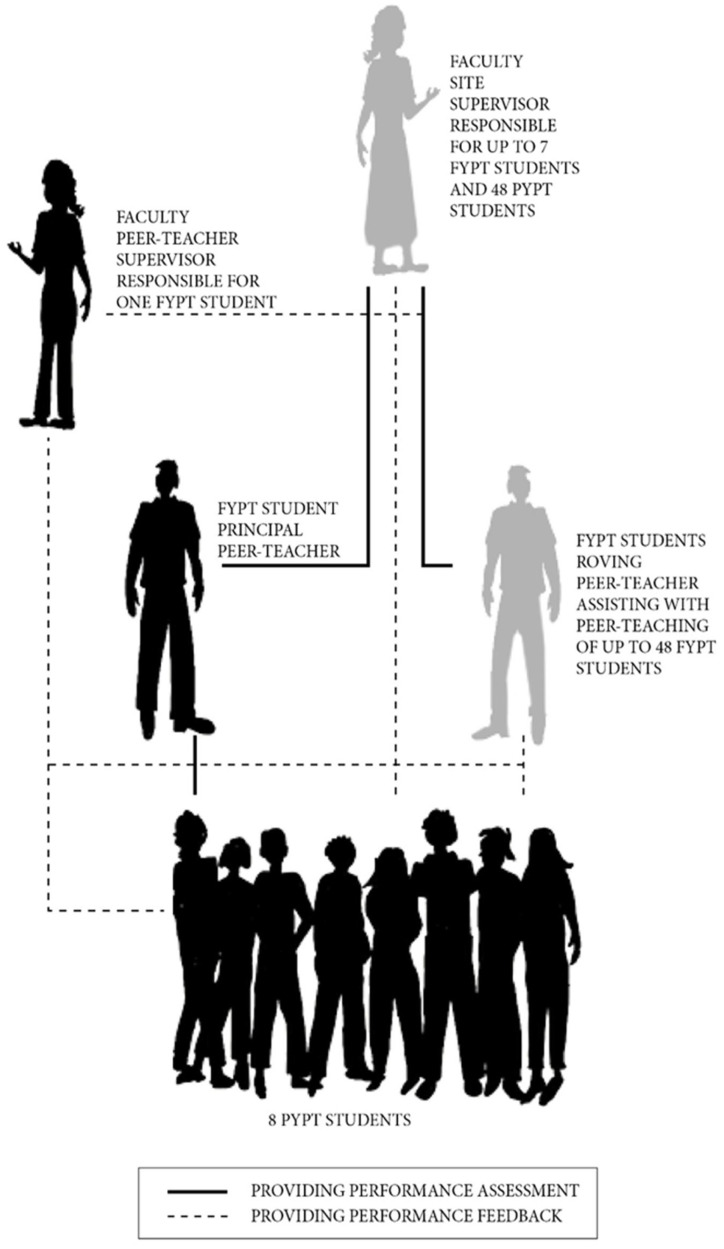
Placement supervision hierarchy.

**Figure 2 ijerph-19-04505-f002:**
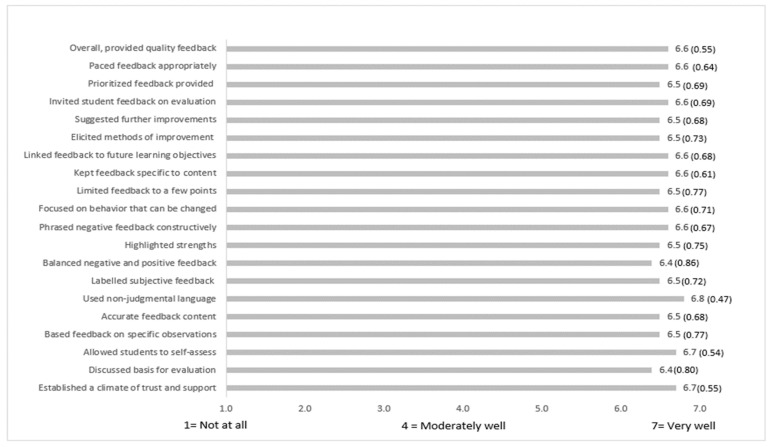
Peer learners’ rating of feedback received from peer teacher, mean (SD), *n* = 170.

**Table 1 ijerph-19-04505-t001:** Relationship between the timing of the PYPT student and FYPT student placement.

Student Group	Placement Weeks Relative to Each Other						
PYPT Group 1, *n* = 93		1	2	3			
PYPT Group 2, *n* = 78					1	2	3
FYPT Group 1, *n* = 13	1	2	3	4	5		
FYPT Group 2, *n* = 11			1	2	3	4	5

**Table 2 ijerph-19-04505-t002:** Positives of being a peer teacher in the model, *n* = 14.

Theme	Subtheme
Personal experience	Enjoyment
	Appreciation
	Confidence
New skillsets learned	Overall collaboration
	Clinical skills
	Professional, organisational and interpersonal skills
Attributes of the teaching model	Relatability
	Approachability
	Mentorship and autonomy
	Opportunity for reflective practice
	Authenticity

**Table 3 ijerph-19-04505-t003:** Challenges of being a peer teacher in the model, *n =* 14.

Theme	Subtheme
Hierarchy	Peer teacher to peer learner
	Pre-existing relationships
Peer teacher-centered issues	Preparation needed
	Responsibility
	Mentoring good students
	Behavioural factors
	Time allocation
Peer learner-centered issues	Lack of buy-in to the peer-assisted learning model
Delivering feedback	Provision of bespoke, succinct specific feedback Difficult conversations

**Table 4 ijerph-19-04505-t004:** Feedback from supervisors of peer teachers, *n =* 16.

Theme	Subtheme
PAL model	Collaborative learning
	Perceived relaxed atmosphere for PYPT students
	Pre-existing relationships
	Insight
	Outcome
	Impact on Faculty
Simulation model	Environments
	Outcomes
Support provided	Training and handover
	Daily support
FYPT students	Enthusiasm, organisation, diligence and professionalism
	Level of clinical acumen
	Level of teaching and communication acumen

## Data Availability

Data sharing is not available.

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
