# Peer review of "Observational Study Exploring the Efficacy and Effectiveness of a New Model of Peer-Assisted Simulation-Based Learning Clinical Placement"

_ijerph, 2022, doi:10.3390/ijerph19084505_

Round 1
Reviewer 1 Report
REVIEW 1: Observational study exploring the efficacy and effectiveness of a new model of peer-assisted simulation-based learning clinical placement
Dear authors,
You have analyzed simulation training versus traditional training in physiotherapy students. Your manuscript is interesting but I need you to answer some questions:
INTRODUCTION
The introduction is very short. The concepts necessary to understand the manuscript are not explained.
The ethical considerations of the research should go in the "material and methods" section.
MATERIALS AND METHODS
Development of the model:
What was the target population? How was the sample chosen? The authors must specify it.
You must explain better the functions of each figure that takes part in the training with simulation. You only say that they provide support but not how they do it.
Implementation of the placement:
What training objectives did PYPT students have as teachers? You must explain how they are trained to be teachers.
How do FYPT evaluate their fellow students? Why is the evaluation not done by the teachers? You justify your answer.
RESULTS
The results are very extensive. You must eliminate redundant or uninteresting information.
REFERENCES
Many bibliographies are obsolete. The bibliographic citations used are more than 5 years old (71.4 %). The authors must update and arrange the bibliography.
Some references are incomplete or have errors. Youshould review this section.
The authors have mixed APA and Vancouver citation regulations. You must write the references correctly.
Author Response
Thank you for your review of our paper “Observational study exploring the efficacy and effectiveness of a new model of peer-assisted simulation-based learning clinical placement”
We appreciate that the feedback provided has helped us make the paper better.
Our responses to comments follow:
INTRODUCTION
The introduction is very short. The concepts necessary to understand the manuscript are not explained. The ethical considerations of the research should go in the "material and methods" section.
Thank you. We agree with hindsight that the background did not provide appropriate detail related to the existing clinical placement, nor the need for the changes we implemented and the outcomes we reviewed. We have heavily edited this section and moved the ethical considerations to the methods.
MATERIALS AND METHODS
Development of the model: What was the target population? How was the sample chosen? The authors must specify it. You must explain better the functions of each figure that takes part in the training with simulation. You only say that they provide support but not how they do it.
Thank you. We hope we have addressed these concerns and provided more detail in this section
Implementation of the placement: What training objectives did PYPT students have as teachers? You must explain how they are trained to be teachers.
Thank you. Learning objectives have now been added in section 2.2.2
How do FYPT evaluate their fellow students? Why is the evaluation not done by the teachers? You justify your answer.
Thank you. As described in lines 154-160, supervisors provided oversight, but as per the PAL model, it was the peers who undertook assessment of the more junior students using the APP (line 154)
RESULTS: The results are very extensive. You must eliminate redundant or uninteresting information.
Thank you. We have reconfigured the way the results are presented and abridged the verbatim quotations and redundant/less relevant information
REFERENCES: Many bibliographies are obsolete. The bibliographic citations used are more than 5 years old (71.4 %). The authors must update and arrange the bibliography. Some references are incomplete or have errors. You should review this section. The authors have mixed APA and Vancouver citation regulations. You must write the references correctly.
Thank you. The list has been reviewed and more recent references have been added to support our statements. We have also fixed the formatting o the references.
Reviewer 2 Report
The topic addressed in this work is of great interest. The number of references used are not enough to support the work presented. In addition, it is necessary to update the works used, since only 2/14 are from 2020 and there are none later. The Method section is quite complete, but I miss the exhaustive description of the sample. The content that corresponds between tables 2 and 3 should be included. The explanation of the Discussion and Limitation is correct, but it is convenient to increase the Conclusions, which are limited to a short paragraph.Author Response
Thank you for your review of our paper “Observational study exploring the efficacy and effectiveness of a new model of peer-assisted simulation-based learning clinical placement”
We appreciate that the feedback provided has helped us make the paper better.
Our responses to comments follow:
The topic addressed in this work is of great interest. The number of references used are not enough to support the work presented. In addition, it is necessary to update the works used, since only 2/14 are from 2020 and there are none later.
Thank you. More recent references have been added to support our statements
The Method section is quite complete, but I miss the exhaustive description of the sample.
Thank you. This has now been added
The content that corresponds between tables 2 and 3 should be included.
Thank you. We have reworked these tables and the accompanying text
The explanation of the Discussion and Limitation is correct, but it is convenient to increase the Conclusions, which are limited to a short paragraph.
Thank you. We have increased our Conclusions section accordingly
Reviewer 3 Report
Review of ijerph-1375587
Observational study exploring the efficacy and effectiveness of a new model of peer-assisted simulation-based learning clinical placement
The manuscript, Observational study exploring the efficacy and effectiveness of a new model of peer-assisted simulation-based learning clinical placement, presents the results of an investigation into the effectiveness of peer-assisted learning (PAL) applied to two levels of students in physiotherapy to move students from being supervised by full-time faculty to being supervised by peer-teachers.
This research is relevant, and it has applications for several fields that require immersive learning. The authors provide some background information (although not enough) about the program and how fieldwork was organized before this study. Coming from a technology background, this reviewer did not understand whether the simulated patients (SPs) were actual computer simulations as in virtual or augmented reality. Not until almost the end of the Results section (p. 12) this reviewer read that the SPs were actors. In the same way, there is not much explanation on other relevant aspects of the setting. For example, lines 55-58 talk about students perceiving inequity because not all of them experienced a “real” setting; does it mean that students wanted to be in the sim or not? What is the setting of the sim? Lines 77-79 state “This was made possible by the collateral development and accessibility of a new simulation facility that could accommodate more than 50 students to undertake simulation-based activities simultaneously” which is still vague. How many hours a day were the students present? How many scenarios were available for students to execute? Did peer-teachers communicate with supervisors by phone (line 113) or video (Webex p. 13)? These seemingly unimportant details paint the picture to understand the context of the study, but they are missing.
The review of literature is scarce in all aspects: the use of sims in the field, the use of PAL, previous results, etc. Basically, the bulk of the literature review can be found between lines 66-72. This reviewer suggests that the article includes more current research that gives validity not only to the setting but also to the results presented. Out of 14 references, only three were published within the past five years.
The authors expressed that they used mixed methods (line 86) for this research. However, the emphasis is on presenting mostly qualitative data. The quantitative data (figure 2, line 220) is quickly explained between lines 201-213 and it is not mentioned again. There is a missed golden opportunity to triangulate quantitative and qualitative data to give validity to each other. That is, the themes that emerged from the reflections coincide a lot with the statements of the scale. It will be interesting to determine if the perceptions expressed in the reflections coincide with the higher or lower scores on the scale. On the other hand, something that the authors did extremely well is the figures because they are attractive and easy to understand.
In terms of qualitative data analysis, the authors presented the themes and some quotes to exemplify them, which is excellent. In the view of this reviewer, there are too many quotes and some of them are repeated to exemplify different themes. Perhaps reducing the number of quotes and creating a table that presents all themes and categories would lend clarity to the manuscript. Peer-teachers are perceived as experts in their field; did overlapping themes emerge from the reflections of peer-teachers and their supervisors? How do they inform the study?
Finally, there is a short discussion on students not needing to be placed in an actual clinical setting to graduate from the program. A discussion about the “inbreeding” of the program is warranted. That is, in future iterations of this research, none of the teachers or learners would have had any experience in a real-world setting. Is this detrimental to the health of the program? What actions can the investigators take to make sure that the program continues to be of quality and change the way the real world is changing?
Author Response
Thank you for your review of our paper “Observational study exploring the efficacy and effectiveness of a new model of peer-assisted simulation-based learning clinical placement”
We appreciate that the feedback provided has helped us make the paper better.
Our responses to comments are below:
This research is relevant, and it has applications for several fields that require immersive learning. The authors provide some background information (although not enough) about the program and how fieldwork was organized before this study.
Thank you. We agree with hindsight that the background did not provide appropriate detail related to the existing clinical placement, nor the need for the changes we implemented and the outcomes we reviewed. We have heavily edited this section
Coming from a technology background, this reviewer did not understand whether the simulated patients (SPs) were actual computer simulations as in virtual or augmented reality. Not until almost the end of the Results section (p. 12) this reviewer read that the SPs were actors.
Thank you. We have made the manner in which SPs were employed more clear and have added in references around evidence based best practice and their use in healthcare professional training.
In the same way, there is not much explanation on other relevant aspects of the setting. For example, lines 55-58 talk about students perceiving inequity because not all of them experienced a “real” setting; does it mean that students wanted to be in the sim or not? What is the setting of the sim?
Thank you. We have made the way the simulated environments mimicked the real environments more clear
Lines 77-79 state “This was made possible by the collateral development and accessibility of a new simulation facility that could accommodate more than 50 students to undertake simulation-based activities simultaneously” which is still vague.
Thank you. This description related to managing the previous lack of workspace, and we have now made this clearer within the text
How many hours a day were the students present? How many scenarios were available for students to execute? Did peer-teachers communicate with supervisors by phone (line 113) or video (Webex p. 13)? These seemingly unimportant details paint the picture to understand the context of the study, but they are missing.
Thank you. This has now been described in more detail under 2.1 Development of the model
The review of literature is scarce in all aspects: the use of sims in the field, the use of PAL, previous results, etc. Basically, the bulk of the literature review can be found between lines 66-72. This reviewer suggests that the article includes more current research that gives validity not only to the setting but also to the results presented. Out of 14 references, only three were published within the past five years.
Thank you. More recent references have been added to support our statements
The authors expressed that they used mixed methods (line 86) for this research. However, the emphasis is on presenting mostly qualitative data. The quantitative data (figure 2, line 220) is quickly explained between lines 201-213 and it is not mentioned again. There is a missed golden opportunity to triangulate quantitative and qualitative data to give validity to each other. That is, the themes that emerged from the reflections coincide a lot with the statements of the scale. It will be interesting to determine if the perceptions expressed in the reflections coincide with the higher or lower scores on the scale.
Thank you. We have now added in some comments about the alignment of qual to quant data.
On the other hand, something that the authors did extremely well is the figures because they are attractive and easy to understand.
Thank you.
In terms of qualitative data analysis, the authors presented the themes and some quotes to exemplify them, which is excellent. In the view of this reviewer, there are too many quotes and some of them are repeated to exemplify different themes. Perhaps reducing the number of quotes and creating a table that presents all themes and categories would lend clarity to the manuscript. Peer-teachers are perceived as experts in their field; did overlapping themes emerge from the reflections of peer-teachers and their supervisors? How do they inform the study?
Thank you. We have reconfigured the way the results are presented in line with this suggestion, and abridged the verbatim quotations and redundant/less relevant information
Finally, there is a short discussion on students not needing to be placed in an actual clinical setting to graduate from the program. A discussion about the “inbreeding” of the program is warranted. That is, in future iterations of this research, none of the teachers or learners would have had any experience in a real-world setting. Is this detrimental to the health of the program? What actions can the investigators take to make sure that the program continues to be of quality and change the way the real world is changing?
Thank you. We think this was misinterpreted, and now have made it clear both in the introduction and the conclusion that this placement in no way replaces the traditional final year placements that are so important to gather ‘real world’ experience. We therefore hope we have addressed this concern.
Round 2
Reviewer 1 Report
The changes suggested by this reviewer have been successfully made